# Long-Term Corrosion of Eutectic Gallium, Indium, and Tin (EGaInSn) Interfacing with Diamond

**DOI:** 10.3390/ma17112683

**Published:** 2024-06-02

**Authors:** Stephan Handschuh-Wang, Tao Wang, Zongyan Zhang, Fucheng Liu, Peigang Han, Xiaorui Liu

**Affiliations:** 1College of New Materials and New Energies, Shenzhen Technology University, Shenzhen 518118, China; zhangzongyan@sztu.edu.cn (Z.Z.); 2210412043@email.szu.edu.cn (F.L.); liuxiaorui@sztu.edu.cn (X.L.); 2Advanced Materials Group Co., Ltd., Fusionopolis Link #06-07, Nexus One-North, Singapore 138543, Singapore; tao.wang1@siat.ac.cn; 3Advanced Energy Storage Technology Center, Shenzhen Institutes of Advanced Technology, Chinese Academy of Sciences, Shenzhen 518055, China

**Keywords:** Galinstan, liquid metal, corrosion, hydrolysis, diamond coating, BDD

## Abstract

Thermal transport is of grave importance in many high-value applications. Heat dissipation can be improved by utilizing liquid metals as thermal interface materials. Yet, liquid metals exhibit corrosivity towards many metals used for heat sinks, such as aluminum, and other electrical devices (i.e., copper). The compatibility of the liquid metal with the heat sink or device material as well as its long-term stability are important performance variables for thermal management systems. Herein, the compatibility of the liquid metal Galinstan, a eutectic alloy of gallium, indium, and tin, with diamond coatings and the stability of the liquid metal in this environment are scrutinized. The liquid metal did not penetrate the diamond coating nor corrode it. However, the liquid metal solidified with the progression of time, starting from the second year. After 4 years of aging, the liquid metal on all samples solidified, which cannot be explained by the dissolution of aluminum from the titanium alloy. In contrast, the solidification arose from oxidation by oxygen, followed by hydrolysis to GaOOH due to the humidity in the air. The hydrolysis led to dealloying, where In and Sn remained an alloy while Ga separated as GaOOH. This hydrolysis has implications for many devices based on gallium alloys and should be considered during the design phase of liquid metal-enabled products.

## 1. Introduction

High-performance computing, high-power light sources, and communication devices, such as wireless communications, are examples of devices necessitating advanced heat dissipation [1]. Heat accumulation in these devices degrades their performance and may shorten their lifespan [2]. To avoid these detriments, heat accumulation needs to be avoided by effective heat transport and heat dissipation. In regard to thermal transport, two materials have shown great performance, namely, diamond and liquid metals [3,4].

Diamond possesses the highest thermal conductivity known, with a thermal conductivity of ca. 2300 W/(mK) [5,6], albeit the thermal conductivity decreases with grain size of the diamond crystals [7]. Besides high thermal conductivity, diamond features thermal, mechanical, and chemical resistance. Furthermore, diamond coatings are biocompatible and do not pollute the environment due to their benign nature. Diamond coatings can be generated using chemical vapor deposition (CVD) techniques, such as hot filament (HF) or microwave plasma (MP) techniques, on non-diamond surfaces. In general, for this deposition, a seeding step is necessary. During seeding, surface defects (in former times) or diamond nuclei are deposited on the substrate surface, which are afterwards grown using the chosen CVD technique [8]. The chosen CVD technique and parameters enable the growth of diamond with varying crystallite sizes, depending on the growth parameters, and varying surface structures, depending on the seeding and initial structure. In regard to structure, a structured diamond coating can also be obtained using etching techniques [9]. Other parameters, such as electrical conductivity, can be controlled by the addition of a boron source in the reactive gas mixture [10]. A deficit of diamond coatings is their roughness necessitating a (soft) thermal interface material (TIM).

Liquid metals are considered interesting candidates for TIMs due to their liquid state (at room temperature) as well as their relatively high thermal conductivity (≈25 W/(m∙K)) compared to other soft thermal interface materials [5], such as thermal greases (0.5–12 W/(m∙K)). Therefore, these liquid metals are already commercialized for thermal interface problems. Further, liquid metals are still investigated for other heat transport problems, such as in magnetocaloric refrigeration for refrigerators and air conditioning [11,12,13]. Liquid metals have also been investigated for a plethora of other thermal transport problems, and improvements in the thermal conductivity of liquid metals have been scrutinized [3]. The thermal conductivity of gallium-based liquid metals can be improved by incorporating high thermal conductivity micro- and nanoparticles, such as graphene sheets and diamond particles [14]. The addition of particles might also alleviate problems of seepage of the liquid metal by increasing the viscosity. Besides thermal conductivity, its other interesting properties are the low vapor pressure (suggesting that it does not solidify when used as a thermal interface material), the low toxicity, the interesting surface chemistry rendering it an interesting environment for chemical reactions [15,16,17,18], and the ability to recycle the liquid metal [19]. Common liquid metals are the eutectic alloy of gallium, indium, and tin (EGaInSn, often denoted Galinstan), which melts at 10.5 °C [20,21]; the eutectic alloy of gallium and indium (EGaIn), which melts at 15.0 °C [20]; and the eutectic alloy of gallium, tin, and zinc (EGaSnZn), melting at 14 °C [22]. A predominant problem of liquid metals is that it is difficulty to handle due to an oxide skin forming in air and the corrosion of most metals by the liquid metal, resulting in the degradation of its thermal conductivity over time or failure of the device.

To alleviate the corrosion issue, corrosion-resistant metals have been put forward, such as Ta [23], W [24], Ti-Ni (and Ni) [25], and Ni-W alloys [26]. Similarly, diamond coatings have been investigated for their compatibility with liquid metals and metal melts. For instance, Naidich showed that most liquid metals do not wet diamond by reactive wetting [27], suggesting the stability of diamond in this regard. Similarly, Handschuh-Wang et al. [28] showed that diamond shows reliable electrical performance during interfacing and a combination of diamond and liquid metal could be used in thermal transport.

This study originates from research on the interfacing of liquid metal with diamond surfaces published previously [28,29]. The corrosion of liquid metals on several metal substrates and substrates protected by non-metal materials have been shown [30,31,32]. In these cases, the liquid metal penetrated the non-metal coating, perhaps by force. Reports on long-term or the high-temperature corrosion of liquid metals have been scarce. This article investigates the corrosion of diamond-protected samples for ca. four years. The working hypothesis was that the liquid metal may penetrate the cavities of the structured diamond coating. As a by-product, this article investigates aging of the liquid metal. Therefore, herein, the stability of liquid metal on different diamond coatings is investigated. At first, the diamond coatings were characterized. Then, the diamond-coated substrates (titanium alloy and silicon samples) were exposed to EGaInSn for four years, stored at an average temperature of ca. 22 °C and average humidity of ca. 70%. The samples were optically investigated each year for a four-year span. Then, the coatings and the “liquid metal” were analyzed to unravel the stability of the liquid metal–diamond system. This analysis was performed by scanning electron microscopy (SEM), energy dispersive spectroscopy (EDS), and mapping as well as X-ray diffraction (XRD) of both the liquid metal and the substrates. Finally, the reason for the solidification of the liquid metal was elucidated.

## 2. Experimental Section

### 2.1. Materials

The eutectic gallium, indium, and tin (EGaInSn, denoted Galinstan) was purchased from Wochang (Dongguang, China). The composition of the alloy was 68.5 wt% Ga, 21.5 wt% In, and 10 wt% Sn. The melting temperature and surface tension are ca. 10.5 °C and ca. 600 mN/m, respectively. The detonation nanodiamond (DND) particles were purchased from Plasmachem GmbH (4 wt%, PL-NanoPure, grade G01, size 10–20 nm, subsequently denoted Plasmachem DNDs) and from Chengdu Dreiway Technology (size 20–40 nm, denoted Chengdu DNDs, Chengdu, China). [2-(methacryloyloxy)ethyl]-trimethylammonium chloride (TMAEMC, 75 wt% in H_2_O) was purchased from Aladdin Industrial Corporation (Shanghai, China). Oxalic acid (≥99.5%) was purchased from Sinopharm Chemical Reagent Co. (Shanghai, China). The Ti-alloy used was a Ti_6_Al_4_V alloy, and the size of the samples was 10 × 10 mm. Concentrated hydrochloric acid was bought from Aladdin. Deionized (DI) water was used as water source unless stated otherwise.

### 2.2. Synthesis of the Diamond Coatings

The synthesis of the diamond coatings can be separated into the seeding step and the growth step. The seeding step was different for the smooth and structured samples. Secondly, there were differences in the diamond growth steps between the nanocrystalline diamond and boron-doped diamond. The synthesis of the diamond coatings is not the main topic of this article and is given below only briefly. Interested readers are referred to previous publications [28,33]. Before seeding, a pretreatment was conducted. Firstly, the Ti-alloy substrates were ground with 800 and 1200 grad SiC sandpaper and were cleaned with water and ethanol by ultrasonically cleaning for 15 min, respectively. The silicon substrates were cleaned and oxidized in a solution of 30% H_2_O_2_, NH_3_·H_2_O, and water (volume ratio 1:1:5) for 10 min (at ca. 80 °C). Then, the samples were cleaned similar to the Ti-alloy sample.

The two seeding approaches: (a) Seeding for a smooth diamond coating was performed following an electrostatic self-assembly seeding approach using Chengdu DNDs stabilized by TMAEMC at a concentration of 5 × 10^−6^ mol/L (for details, see also reference [34]). The DND concentration was diluted to a final wt% of 0.005 and adjusted to a pH of 3. The substrates were immersed in the seeding solution and seeded ultrasonically for 30 min. (b) Seeding for a structured diamond coating was performed following an electrostatic self-assembly seeding approach using poorly stabilized Plasmachem DNDs. The Plasmachem DNDs were stabilized with 7 × 10^−5^ mol/L oxalic acid. The dispersion was adjusted to a pH of 5. The samples were ultrasonically seeded in this solution for 30 min. Subsequently, both samples were carefully dried by blowing with nitrogen. The dried samples were ready for deposition of the diamond coatings by hot filament chemical vapor deposition (HFCVD).

Nanocrystalline diamond deposition: Irrespective of the structure on the surface, the growth of the diamond was carried out with the same parameters. The flow rates of the reaction gases methane and hydrogen were 32 and 800 sccm, respectively. Further, the temperature of the hot filaments was adjusted to ca. 2500 °C and the pressure was set to 1.5 kPa. The deposition was conducted for 1 h, resulting in either a smooth nanocrystalline diamond coating (for the high seeding density) or growth of hemispheres on the substrate (low seeding density). The synthesis of the boron-doped diamond (BDD) was executed similar to the nanocrystalline diamond. To obtain boron doping, a boron source (trimethyl borane, TMB) was added into the gas mixture, in the form of a gas mixture (0.1% TMB, 99.9% hydrogen). The flow rates of the three gases methane, hydrogen, and boron source were 32 sccm, 400 sccm and 160 sccm, respectively. The deposition was carried out for 1 h at a pressure of 2 kPa and a filament temperature of 2500 °C. These parameters typically yield in microcrystalline BDD.

### 2.3. Aging/Corrosion Experiment of the Diamond/Liquid Metal Systems

The samples were cleaned before use by water and ethanol. Then, the liquid metal (1–2 g, depending on size of the sample) was deposited on each sample. Subsequently, the liquid metal was forced to wet the sample (forced wetting) [35]. The samples were placed in separate closed petri dishes closed with a lid (not air-tight) and stored in a dark drawer for 4 years. The average temperature in the summer months (6 months) was ca. 25 °C at a humidity of ca. 80%. The average temperature in the winter months was ca. 18 °C at a humidity of ca. 60% or less (these values somewhat correlate to the climate in south China). Every year, the samples were visually inspected and changes were noted. After 4 years, the “liquid metal was removed” by using a plastic tweezer and transferred into another petri dish. The surface of the samples was superficially cleaned by water, 1 min immersion in 1 mol/L HCl and water, successively.

Samples: On several samples, the liquid metal was deposited, as detailed in Table 1. The main samples discussed in this article are Samples 2, 3, 8, and 9. However, the liquid metal solidified on all of the samples during the time of observation (4 years).

### 2.4. Characterization

Scanning electron microscopy (SEM) micrographs were taken with the GeminiSEM 300 (Carl Zeiss Microscopy Ltd.; Cambridge, UK) at an acceleration voltage of 15 kV. Energy-dispersive X-ray spectroscopy (EDS) was measured with the abovementioned SEM at an acceleration voltage of 20 keV. Raman spectra were measured with the LabRAM Odyssey (Horiba, Kyoto, Japan). The incidence laser wavelength was 532 nm. The X-ray diffraction (XRD) pattern was determined with the Empyrean (Malvern Panalytical, Eindhoven, The Netherlands) with Cu Kα1 radiation (λ = 0.154 nm). The cross-section of the diamond coated silicon sample was obtained by cutting the silicon with a diamond pen cutter, followed by breaking the substrate. Photographs were taken with a private camera (Canon EOS 700 D) with a Canon EF-S 18–135 mm lens.

## 3. Results

### 3.1. Surface Morphology and Structure of Diamond Coatings after Contact with Liquid Metal

Figure 1 shows the surface morphology of the coatings after 4 years of contact with Galinstan at room temperature. The coatings were either flat (smooth; Figure 1a; later, by Raman spectroscopy, discovered to be graphite) or structured diamond coatings (Figure 1b–f). Further, the diamond coatings were either nanocrystalline (Figure 1b,d,e) or sub-microcrystalline boron-doped diamond (Figure 1c,f). Crystal facets are difficult to discern here, which hints that the coating in Figure 1a is not diamond, rather a graphitic phase (see paragraph, Raman). Figure 1b,d,e show structured nanocrystalline diamonds coated on titanium alloy (two different samples). The diamond spheres are approximately 2.3 ± 0.4 µm in diameter and form a multilayer on the titanium alloy substrate. The structure is consistent with the one published previously [28], namely, a diamond sphere size of ca. 2.5 µm and a similar multilayer structure of the coating. In a previous article, short-term interfacing of Galinstan with non-conductive diamond and conductive BDD coatings was investigated [28] and found to be satisfactory. Figure 1e shows a magnification of the sample in b and provides a rough estimate of the crystallite size, which is ca. 20 nm [33]. The structure of this coating originated from the seeding procedure and the growth mechanism of diamond with CVD. The seeding was conducted with nanodiamond seeds, which show a low but controlled adsorption to the Ti substrate. This was achieved by adjusting the surfactant, pH, concentration of particles, and zetapotential of the particles [33]. Depending on the growth environment, either spherical (Figure 1b) or hemispheres (Figure 1c) are grown on the substrate due to the island growth mechanism of CVD diamond. The more coalesced coating formed in Figure 1c can be attributed to a higher seeding density (different substrate). The crystallite size of this diamond coating is with ca. 500 nm considerably bigger than the nanocrystalline diamond coating in Figure 1b,d and is called hereforth sub-microcrystalline.

After contact with the liquid metal for four years, the four different coatings from Figure 1 were subjected to Raman spectroscopy to determine the presence of diamond, and the spectra are given in Figure 2. Sample 2 in this figure corresponds to the image shown in Figure 1a, the smooth graphite coating on Si. Three broad peaks were detected at 1349 cm^−1^, 1580 cm^−1^, and 2698 cm^−1^. No peak at 1332 cm^−1^ was observed signifying the absence of diamond. The peaks at 1349 cm^−1^ and 1580 cm^−1^ denote the graphite D and G bands, respectively [36]. The Raman spectra of sample 3 and sample 9 are similar. These samples are the structured nanocrystalline diamond on titanium shown in Figure 1b,d,e. The Raman spectra show peaks at ca. 1335 cm^−1^ (1333 cm^−1^) and 1582 cm^−1^. The band at 1582 cm^−1^ is the G-band of graphite, signifying the presence of some graphite as a by-product of the nanocrystalline diamond. The peak size is exaggerated compared to the other band due to the higher sensitivity of that band compared to the diamond band. The diamond peak is typically located at 1332 cm^−1^ [37]. In our samples, this band is shifted toward higher wave numbers, a direct result of compressive stress (−1.58 GPa for 1335 cm^−1^) [38] originating from the mismatch in the coefficient of thermal expansion [39] between the titanium alloy and the diamond coating. Further, a shoulder at 1151 cm^−1^ and a minute peak at 1460 cm^−1^ (for sample 9) can be discerned, which stem from the presence of transpolyacetylene (t-PA) [40,41]. Sample 8 is boron-doped diamond on Si; the corresponding SEM micrograph is shown in Figure 1c,f. The diamond band at 1332 cm^−1^ as well as the graphite bands (i.e., 1580 cm^−1^) cannot be observed here. A peak at 1201 cm^−1^ and a shoulder at 1288 cm^−1^ have been detected, signifying the generation of BDD. The sample was made in the same batch like the coated sample in our previous publication [28]. According to the peak shift method, the boron concentration was determined to around 2.8 × 10^21^ cm^−3^ [28,42]. This concentration is higher than the threshold concentration, resulting in good conductivity of the BDD [43,44]. Comparing the Raman spectra and the surface morphology of the samples after contact with the liquid metal with the previous published results of as-grown diamond coatings [28], no significant change in morphology and Raman spectra can be discerned, illustrating that the diamond coatings were not corroded.

### 3.2. Corrosion Resistance and Penetration of the Diamond Coatings by the Liquid Metal

The corrosion tests were performed on several diamond coated titanium alloy samples as well as diamond-coated Si samples. As a control experiment, the liquid metal (EGaInSn) was also deposited on unprotected titanium alloy. The liquid metal was forced to wet the samples according to the description in the experimental section and in a previous publication [35]. A photograph of the samples before deposition (Figure 3a) and after liquid metal deposition on the substrates (Figure 3b) signifies wetting of the liquid metal on the substrate. The sample numbers in the image correspond to the sample number given in the Raman discussion.

The liquid metal samples were sporadically (every year) investigated by eye. In the first year, no change was observed. After storage for two years, the liquid metal on three samples became sluggish, namely, sample 3 (one of the two rough diamond on Ti samples), sample 4 (one of the two bare Ti samples), and sample 8 (one of the rough BDD on Si samples), while the other samples remained liquid. It became more solid (or putty)-like. The exterior changed from a silvery dull color to partially dark grey. After four years, all liquid metal samples on the coatings were partially solidified. Upon indenting the former liquid metal, a grey material could be observed, as shown in Figure 4a. The liquid metal behaved similar to a liquid metal putty. Liquid metal putties are mixtures of liquid metal with micro- or nanoparticles, endowing them with more solid-like behavior and enabling molding the material in shapes [45]. At the same time, another part of the liquid metal remained partially liquid with a silvery sheen and liquidity, as shown in Figure 4b. At this stage, the hypothesis was that the liquid metal somehow got into contact with the titanium alloy and dissolved aluminum. It is known that gallium corrodes aluminum and aluminum alloys (albeit aluminum has limited solubility in gallium) [46,47]. This corrosion commences via the diffusion of gallium into the grain boundaries and the dissolution of aluminum [46]. This corrosion is an issue for aluminum heat sinks in contact with gallium but is useful for hydrogen production via the reaction of aluminum with water, where gallium activates the aluminum [46]. Further, dissolved aluminum oxidizes on the liquid metal surface, resulting in a dark grey to black oxide on the surface. Therefore, an assumption was that alloying with aluminum from the Ti alloy lead to the solidification of the liquid metal. Disappearing (likely drying) was observed previously [48,49] for liquid metal bearings (gallium alloy in contact with copper) presumably due to alloying. Contradicting this hypothesis, the liquid metal also solidified on the diamond coatings on Si, as no metal for alloying is available on the Si-coated samples.

In order to rule out diffusion of liquid metal across the diamond coatings toward the substrates, SEM and EDS cross-sections were measured. Further, an EDS spectrum was taken from an unprotected titanium alloy sample after being in contact with the liquid metal for four years (Figure 5a). This EDS spectrum shows several features that are peaks at 0.45 and 4.5 keV, denoting the titanium Lα and Kα bands, respectively. Further, peaks at 1.49 and 4.949 keV denoting the presence of aluminum and vanadium, respectively, were detected. Finally, a weak peak at 1.098 keV was detected, indicating the presence of gallium in the alloy. This is a bit surprising, as the liquid metal was taken off the sample, and the sample cleaned briefly by acid. Indeed, Mingear and Hartl have shown that Ti can withstand corrosion by gallium alloys [25]. However, they did not perform a long-term experiment, but only exposure to liquid metal at 220 °C for 300 h. Further, they used a pure Ti substrate, while in this experiment, a titanium, aluminum, and vanadium alloy is used, and aluminum is known to be soluble in gallium alloys. After inspecting the EDS spectra of the liquid metal, which was deposited on the titanium alloy for four years, the main peaks are that of gallium at 1.1 keV (Lα) and oxygen at 0.53 keV (Kα). A marginal peak in the spectrum from aluminum at ca. 1.5 keV can be discerned (see Appendix A, supporting information). Further, peaks for the elements indium and tin are very weak. The peaks for indium should be located at 3.29 keV, 3.5 keV, and 3.71 keV, denoting In Lα, In Lβ1, and In Lβ2 [20]. Further, the peaks for tin should be located at 3.44 keV (Sn Lα), 3.66 keV (Sn Lβ1), and 3.90 keV [20], albeit the first two overlap with the peaks of indium and thus cannot be discerned. The amount of aluminum in the EGaInSn alloy was assumed too low to be able to explain the solidification of the liquid metal at this point. The main composition was assigned, at first, to gallium oxide. One can also see already that the In and Sn peaks are much lower than expected [20]. In Section 3.3, it is shown that the liquid metal at this location is GaOOH and not gallium oxide.

Still, the investigation of the resistance of the diamond coatings toward the diffusion and corrosion was continued. In Figure 5b, the cross-section of the boron-doped diamond-coated Si samples is shown. The upper material is the boron-doped diamond with a thickness of ca. 3.5 µm. Below this diamond coating, one can see the substrate material and the good interfacial bonding of the coating, resulting in good adhesion (as no cavities can be seen). In Figure 5c, the corresponding cross-sectional EDS line scan is shown, where, C, Si, Ga, In, and Sn were detected along the line shown in Figure 5b. The EDS line scan detects, at first, carbon from the diamond coating (note that boron cannot be detected with EDS reliably) with a cps of ca. 3000. At a depth of 3.5 µm, the C concentration quickly dwindles and Si is detected. There is a coexistence region between 3.5 and 4 µm, where carbon and Si coexist, stemming from SiC formation during the growth process, which is a reason for the good adhesion of diamond coatings on Si. Then, the cps for Si stay constant. In the whole line scan, no Ga, In, or Sn can be detected, indicating that the boron-doped diamond coating can effectively protect against the diffusion of the liquid metal towards the substrate surface. A similar result was obtained for the EDS line scan of the diamond-coated titanium alloy, as shown in Appendix A, supporting information. The EDS line scan signifies the presence of carbon at the top and titanium in the bottom layer. No contamination of the titanium alloy with Ga, In, or Sn could be detected with the EDS measurement.

The inability of the liquid metal to penetrate the diamond coating is a promising outcome, especially as it can substitute as a conductive BDD coating corrosion-resistant structure metals, such as Mo and Ta, for electrical contacts. One should note that the experiment was conducted at room temperature. As the combination of diamond and liquid metal is anticipated for cooling/heat spreading applications, the reader should note that diamond is stable against gallium at considerable temperatures [27,50], while metals (and alloys) will be impacted by a threshold temperature at which corrosion, dissolution, and/or intermetallic formation take place.

### 3.3. The Origin of the Solidification of the Liquid Metal?

The dissolution of aluminum was ruled out as the reason for the solidification of the liquid metal in the previous section because of several reasons: (1) The solidification took place on all samples, namely, titanium alloy samples, titanium alloy samples protected by diamond, and Si samples protected by diamond/graphite. The latter does not have aluminum as a constituent. (2) Dissolved aluminum in the liquid metal was barely detectable in the EDS spectrum of liquid metal stored on bare titanium and could not be detected in the EDS spectra of the liquid metal on the diamond-coated Ti and Si samples (see Appendix A).

Appendix A show something surprising. The liquid metal in these figures is not smooth and homogeneous. Rather, it features smooth regions and regions with features which appear to be crystal-like. Therefore, EDS spectra of these different regions were taken (see supporting information, Appendix A). Appendix A is from a smooth region of the liquid metal. Here, the signal for Ga (1.1 keV, Ga Lα) is clearly visible. Similarly, clear signals for In (3.29 keV, 3.5 keV, and 3.71 keV) and Sn (3.90 keV) can be observed. The oxygen signal at 0.53 keV is much smaller compared to the signals for indium and tin. The strong signals for indium and tin suggest a higher concentration of these components at this location of the alloy than in the pure and fresh liquid metal (compared to the same batch of the alloy in ref [20]). Notably, the EDS spectrum also shows the presence of Si (1.75 keV, Kα), which is the substrate of this sample. Appendix A shows the EDS spectrum of the rough region of the liquid metal. The spectrum features prominent peaks for O (0.53 keV) and Ga (1.1 keV) and weak peaks for Si (1.75 keV), In (3.29 keV, 3.5 keV, and 3.71 keV) and Sn (3.90 keV). The Si peak originated from the underlying substrate. Strikingly, the In and Sn peaks are very weak compared to the EDS spectrum at the smooth liquid metal location, suggesting a lower concentration of In and Sn at this location. This separation of the liquid metal is seen for EGaInSn when it transforms from gallium oxide to gallium oxide hydroxide in water (due to hydrolysis) [51], and the process is called hydrolysis-induced dealloying. This hydrolysis is known to commence slowly in water and is accelerated by heat [52]. The formation of gallium oxide hydroxide at hydro-thermal conditions has long been known (typically, in steam or in water) [53]. This change can be a reason for the change in the liquid metal appearance and its hardening. Therefore, the EDS spectra suggest hydrolysis and dealloying of the liquid metal.

To clarify the existence of GaOOH and the associated dealloying, an XRD diffraction pattern and EDS mapping were captured. The XRD pattern collected from the solidified liquid metal shown in Figure 6 clearly identifies the presence of GaOOH. The agreement with PDF #54-0910 is striking. The strongest reflections in the pattern were observed for 2θ = 21.4, 33.6, 35.8, 53.8, and 59.9°, representing the crystal facets (110), (130), (021), (111), (221), and (151), respectively [54]. Further, all other diffraction peaks for GaOOH have been observed. Notably, for 2θ values between 30 and 40°, an underlying broad diffraction peak is observed. This broad peak originates from the liquid metal (without hydrolysis). In a fully crystalline sample, the broad peak vanishes [54]. Compared to previous research on Ga nanoparticles in water [52], this peak is much weaker, suggesting a higher conversion of the liquid metal (liquid) toward a solid (GaOOH).

In the previous paragraphs, the formation of GaOOH was proven and dealloying was suggested as a by-product of the continuous hydrolysis of gallium oxide. Dealloying was already suggested by the EDS spectra of the aged liquid metal in Section 3.2. To prove dealloying of the liquid metal, EDS maps were taken. An SEM image (to view the morphology) together with the EDS maps and the EDS spectrum is shown in Figure 7. The SEM micrograph shows a somewhat rugged surface (compare the smooth liquid metal surface shown in Appendix A) with a circular arrangement of solid-like features. The accompanying EDS maps show the location of Ga, In, Sn, and O. The images for Ga and O are bright at the same locations, associating these two elements with each other (one can assume either gallium oxide or GaOOH). However, there is one exception. An area to the right side features the presence of Ga, while little O is present at the same location. Our assumption is that this region is rich with un-hydrolyzed (but oxidized) liquid metal. In this region, In and Sn are homogeneously distributed and their signals are strong. In contrast, in other regions, the distribution and intensity of In and Sn are inhomogeneous, maybe best seen in the lower middle region, where little In and Sn can be observed, while Ga and O EDS maps signify the presence of these elements. Therefore, we conclude that the liquid metal is dealloyed into an InSn rich alloy and GaOOH crystals. The dealloying (higher In and Sn concentrations) increases the melting temperature of the residual “liquid metal”, which, at some point, solidifies. The origin for this dealloying is the continuous hydrolysis in air (due to water vapor/humidity and accelerated by heat). This hydrolysis commences even though the oxide skin is often suggested to be rather unreactive and suggested to be limiting further oxidation [55]. The GaOOH crystals, on the other hand, continue to grow with time, as seen in our previous article [52]. Currently, the speed of this hydrolysis in air is unknown. Presumably, hydrolysis in air is accelerated by temperature and increased humidity, with an unknown onset humidity. Other factors affecting hydrolysis speed could be the salt content in the air (for regions close to the ocean), and all these effects should be scrutinized in detail.

Hydrolysis and phase separation result in the solidification of the liquid metal during aging and are detrimental to their use in thermal interface materials (due to heat accumulation). These processes are very important for all nanoscale and microscale devices employing gallium-based liquid metals. This importance is in regard to the long-term stability, ability to store them, and reliability. For instance, gallium-based liquid metal micro- and nanodroplet-based devices and catalysts (such as in gallium-rich supported catalytically active liquid metal solutions, SCALMS) [56] will inevitably hydrolyze in air, which reduces the overall lifetime in these applications and has implications for storage. This may be similar to iron (II) solution, which needs to be prepared prior to use as oxygen in air oxidizes it to iron (III). Further, this has implications in the field of thermal transport, as the liquid metal is exposed to elevated temperatures, and the resulting drying leads to deteriorating thermal transport. Drying of liquid metal for consumer CPUs or GPUs has been discussed in forums, albeit the posters assign this effect often to “oxidation” [57]. Cleaning the area of the thermal transport and reapplying this is fairly simple for a consumer, albeit reapplication may lead to spillage of liquid metal (which may destroy the main board). In applications related to (wireless) communication, this drying effect is detrimental, as these devices are expected to run for around 10 years without much maintenance [58]. In other applications, the service life should also be improved, such as thermal interface materials for CPUs and GPUs [59]. Similarly, the hydrolysis is also a challenge for other applications, such as stretchable electronics (especially, miniaturized electronics), as the electrical conductivity is far lower than that of the liquid metal [5] and could lead to discontinuity due to its brittle and powdery nature (see Figure 4a). Liquid metal seals and liquid metal bearings (bearings for computed tomography units) are similarly impacted.

An inherent issue in regard to hydrolysis is the fact that it is challenging to avoid hydrolysis as water vapor diffusion across elastomers is rather fast, resulting in poor protection versus the hydrolysis and associated solidification. In this regard, perhaps metal seals feature improved barrier properties but are difficult to implement in a cost-efficient manner. Vacuum conditions (or protective atmosphere) and antioxidants are also possible means to circumvent the issue but are similarly difficult to implement; exceptions are the liquid metal thermometers and sphygmomanometers. A reduction in the humidity (de-humidifier) in the environment appears to be the easiest method to mitigate the hydrolysis, at least for network clusters.

## 4. Conclusions

In a long-term experiment over 4 years, the liquid metal Galinstan was brought into contact with various diamond coatings. These diamond coatings were used to serve as a protective layer against the liquid metal corrosion of an underlying substrate. The substrate (metal) was protected against direct contact of the liquid metal, avoiding liquid metal corrosion or embrittlement. However, the liquid metal solidified over time and this solidification was not correlated with the underlying substrate, albeit some interaction of the liquid metal with the unprotected titanium substrate could be observed. Rather, the solidification was traced to 1. oxidation of the liquid metal, followed by 2. hydrolysis of the liquid metal toward GaOOH, proven by XRD, and this hydrolysis was accompanied by dealloying into GaOOH crystallites and an InSn-rich alloy, as evidenced by EDS. These processes affect applications ranging from thermal interface materials, stretchable electronics, slip rings, and switches via catalysis and solvents to biomedical applications of liquid metals based on liquid metals and is exacerbated at high temperatures, high humidity, and small liquid metal entity sizes. Researchers and engineers should be aware of these processes resulting in the solidification of liquid metal over time and design a sufficient protective environment to avoid this to improve the lifetime of thermal dissipation systems and other devices based on liquid metals.

## Figures and Tables

**Figure 1 materials-17-02683-f001:**
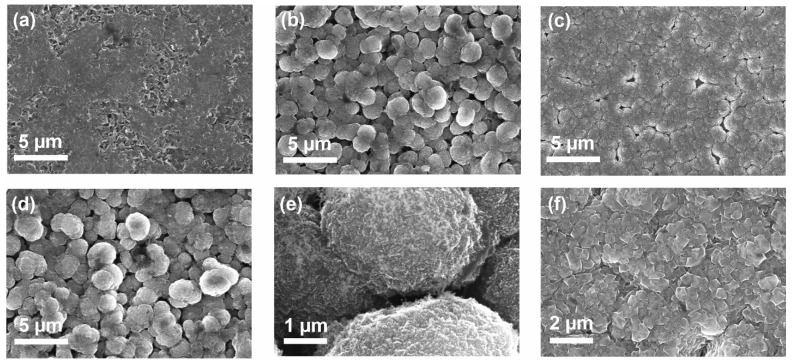
SEM micrographs of the diamond coatings after interfacing with liquid metal for four years. (**a**) Graphitic phase grown on Si. (**b**) Structured nanocrystalline diamond on Ti. (**c**) Structured boron-doped (sub-microcrystalline) diamond on Si. (**d**) Structured nanocrystalline diamond on Ti. (**e**) and (**f**) are magnifications of (**c**) and (**d**), respectively.

**Figure 2 materials-17-02683-f002:**
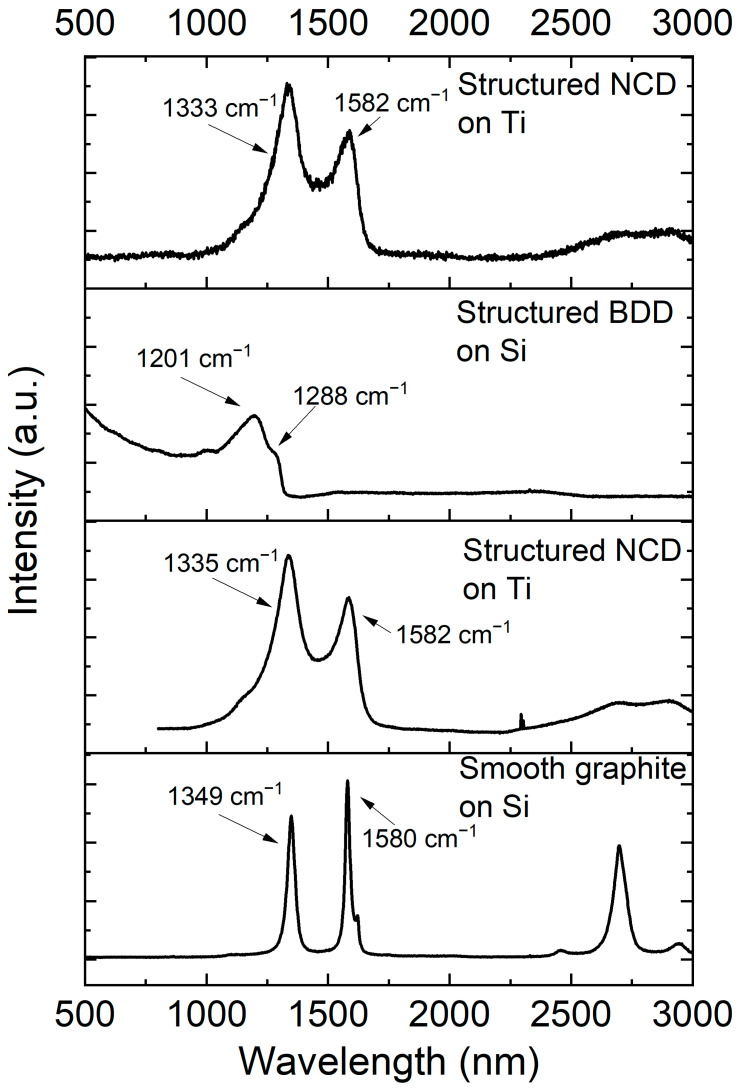
Raman spectra of the diamond coatings on Ti and Si after interfacing with the liquid metal for four years. The samples are flat graphite on Si, structured diamond on Ti, structured BDD on Si, and structured diamond on Ti.

**Figure 3 materials-17-02683-f003:**
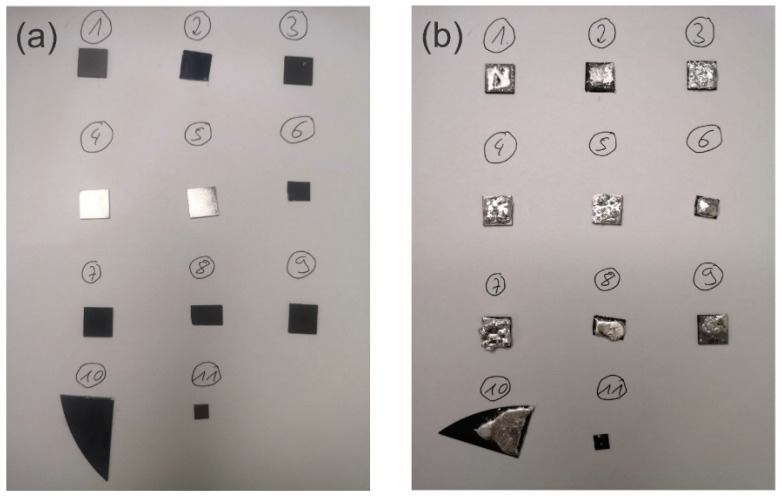
Samples (**a**) before and (**b**) after deposition of liquid metal and forced wetting. (1) Flat diamond on Ti, (2) flat graphitic phase on Si, (3) rough diamond on Ti, (4, 5) bare Ti, (6) BDD on Si, (7) rough diamond on Ti, (8) rough BDD on Si, (9) rough diamond on Ti, (10) BDD on Si, and (11) BDD on Ti.

**Figure 4 materials-17-02683-f004:**
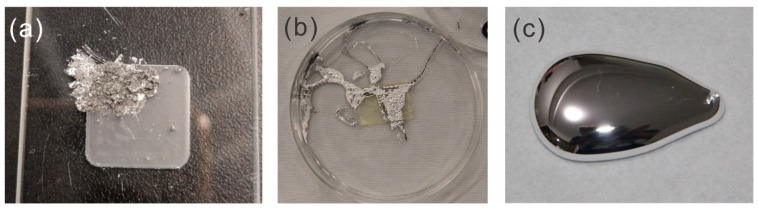
Liquid metal after aging for 4 years on the diamond-coated Ti sample. (**a**) Solidified part of the liquid metal (formerly ca. 200 µL). (**b**) Semi-liquid part of the liquid metal; petri dish diameter: 10 cm. (**c**) Liquid metal before aging in this application (ca. 1500 µL). The liquid metal was always Galinstan.

**Figure 5 materials-17-02683-f005:**
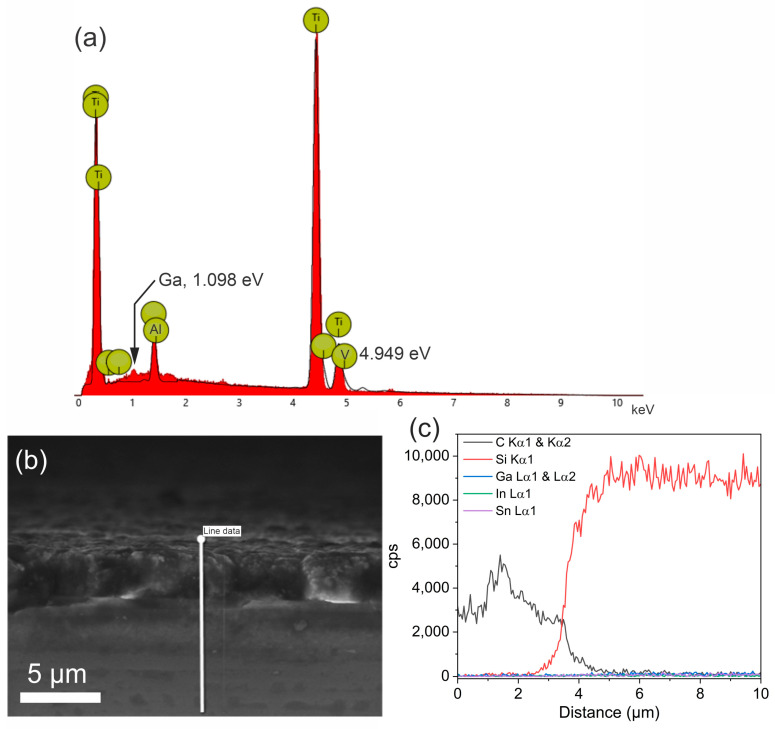
(**a**) EDS spectrum of the Ti sample after being in contact with Galinstan for 4 years followed by cleaning with acid for 1 min. (**b**) SEM cross-section of boron-doped diamond-coated Si after contact with liquid metal for 4 years. The liquid metal was removed by cleaning with acid for 1 min. Then, the sample was cut and the new cross-section measured. The line indicates the direction of the EDS line scan. (**c**) EDS line scan of (**b**) showing the counts per second of C, Si, Ga, In, and Sn.

**Figure 6 materials-17-02683-f006:**
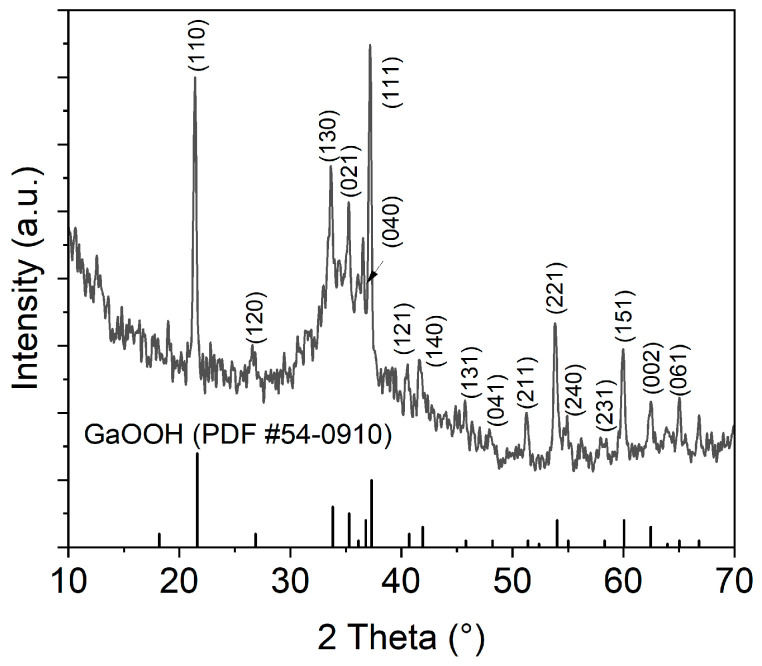
XRD spectrum of the solidified part of the liquid metal. The spectrum relates to the image given in the previous Figure 5a.

**Figure 7 materials-17-02683-f007:**
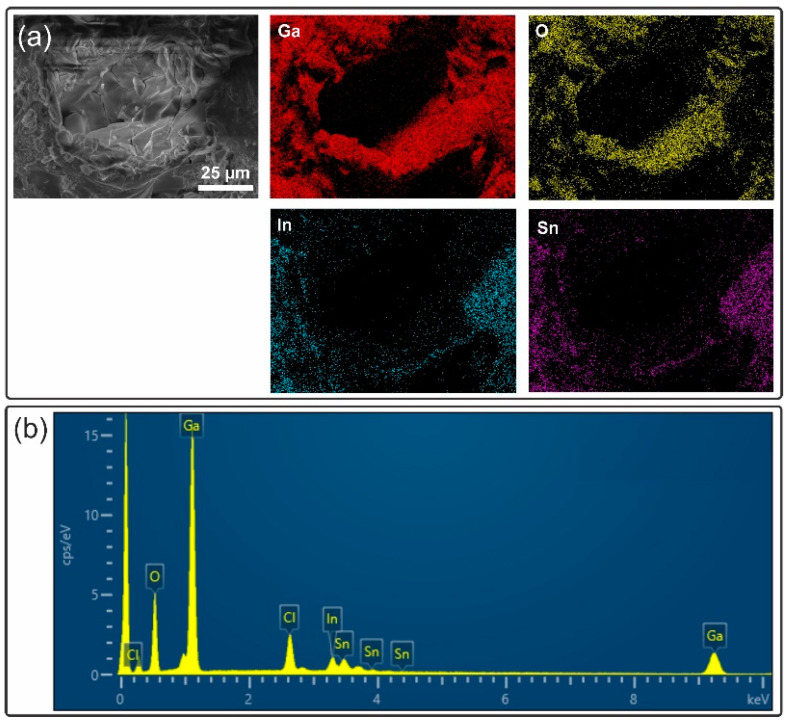
Dealloying of the liquid metal. (**a**) SEM and EDS mapping of the solidified liquid metal after aging for 4 years on diamond-coated Ti. (**b**) EDS spectrum of the aged and solidified liquid metal.

**Table 1 materials-17-02683-t001:** List of all samples made.

Sample ID	Substrate	Diamond Coating
#1	Titanium	Smooth nanocrystalline diamond
#2	Si	Smooth graphite
#3	Titanium	Structured nanocrystalline diamond
#4	Titanium	Uncoated
#5	Titanium	Uncoated
#6	Si	Smooth BDD
#7	Titanium	Structured nanocrystalline diamond
#8	Si	Structured sub-microcrystalline BDD
#9	Titanium	Structured nanocrystalline diamond
#10	Si	BDD
#11	Titanium	BDD

## Data Availability

Data available on request from the authors.

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
