# Peer review of "Long-Term Corrosion of Eutectic Gallium, Indium, and Tin (EGaInSn) Interfacing with Diamond"

_materials, 2024, doi:10.3390/ma17112683_

Round 1
Reviewer 1 Report
Comments and Suggestions for Authors
Manuscript by Stephan Handschuh-Wang et al. reports the investigation into the long-term corrosion effects of eutectic gallium, indium, and tin (EGaInSn) interfacing with diamond surfaces. The study is notably well-structured and meticulously detailed in its approach, delivering valuable insights into the stability and compatibility of diamond coatings with liquid metals for thermal management systems. The authors have demonstrated the resistance of diamond coatings to corrosion and penetration by liquid metals over an extended period, a significant finding considering the corrosivity of liquid metals like EGaInSn towards conventional metals used in electronic devices. The study elucidates the phenomenon of liquid metal solidification due to oxidation and subsequent hydrolysis to GaOOH, a process that was verified through various analytical techniques, including SEM, EDS, and XRD.
· The manuscript mentions that the diamond did not corrode nor was it penetrated by the liquid metal. Could you elaborate on whether any microstructural changes or phase transformations occurred within the diamond coating itself over the long period of exposure to EGaInSn?
· The report outlines the hydrolysis of gallium alloys in detail. Given the significant role of environmental humidity in this process, how might varying humidity levels affect the rate and extent of liquid metal solidification and the overall stability of the system?
· In your experimental setup, was there any consideration or measurement of mechanical stresses that might have arisen due to the differing thermal expansion coefficients of the metal substrate and the diamond coating, particularly under temperature variations?
I recommend this manuscript for publication in your materials. However, addressing the above questions should be completed before considering it for publication.
Comments on the Quality of English Language
Recommended to check for the typos
Reviewer 2 Report
Comments and Suggestions for Authors
The work needs to be corrected in several aspects as described below.
1. The authors wrote: ‘The diamond coatings were either flat (smooth, Figure 1a, later discovered to be graphite)’ (see lines 178-179). Although in the discussion of Figure 1a, there is a note about the later discovery that the coating was graphite, but the clarification of this discovery process should be added to the text.
2. The authors wrote” “The structure is consistent with the one published previously.28” (see lines 185-186). Although the text mentions that the structure is consistent with previously published work (reference 28), in my opinion authors should briefly summarise the key findings of that work to provide context.
3. In my opinion, authors should add more explanation between the experimental observations and the hypotheses in Section 3.2., especially of the liquid metal solidification process, referencing similar phenomena in other studies for comparison. Moreover, the hypothesis of aluminium dissolution should be discussed with more references to the existing literature on gallium alloy interactions with various metals.
4. The information presented in lines 238-246 should be described in more detail. The authors should clarify the timeline of observations and the transition from liquid to solidified states of the liquid metal, as well as expand on the reasons why the initial hypothesis about aluminium dissolution was ruled out.
5. In Section 3.3 authors explained the solidification process and the role of hydrolysis and dealloying, however, additional context on the relevance of these findings to broader applications, such as in thermal interface materials and electronics, would enhance the discussion.
6. The authors should add a discussion, based on the results obtained, about potential strategies to mitigate the hydrolysis and dealloying effects in practical applications.
7. The conclusion section should be expanded. For example, while the process of oxidation and hydrolysis leading to the formation of GaOOH and dealloying is mentioned, more details on how these findings affect practical applications should be included. Furthermore, the authors should discuss the broader implications for the use of liquid metals in various technologies and the importance of developing protective measures.
8. 8 references by one author (Stephan Handschuh-Wang) is a lot for 54 references (above 14%). The authors should minimize this or justify the need for citation.
9. The authors should standardize the work in terms of the style of the manuscript. For example, in many places there is an incorrect font.
Comments on the Quality of English LanguageThe English needs minor corrections; for example
Instead of: "The signal for oxygen at 0.53 keV is very small compared to the signals denoting indium and tin." is better to use: "The oxygen signal at 0.53 keV is much smaller compared to the signals for indium and tin."
Round 2
Reviewer 2 Report
Comments and Suggestions for Authors
The authors have improved the manuscript considerably. I have no further comments